# Extended Prescribing Roles for Pharmacists in Poland—A Survey Study

**DOI:** 10.3390/ijerph19031648

**Published:** 2022-01-31

**Authors:** Jagoda Miszewska, Natalia Wrzosek, Agnieszka Zimmermann

**Affiliations:** 1Students’ Scientific Circle of Pharmaceutical Law, Department of Medical and Pharmaceutical Law, Medical University of Gdansk, 80-210 Gdansk, Poland; jagodam@gumed.edu.pl; 2Department of Medical and Pharmacy Law, Medical University of Gdansk, 80-210 Gdansk, Poland; natalia.wrzosek@gumed.edu.pl

**Keywords:** pharmacy prescribing, pharmacy legislation, entitlements of pharmacists

## Abstract

Background: In recent years, a systematic increase in the role and powers of pharmacists has been observed. The COVID-19 pandemic has shown that this is a professional group that is extremely necessary for the smooth functioning of the health care system. One of the important powers of pharmacists is the possibility of issuing prescriptions for both patients in the pharmacy and for themselves and their family members. Polish pharmacists obtained extended entitlements in this field in March 2020. Due to the extension of pharmacists’ prescribing rights in Poland, it was decided in this study to determine the current practice of pharmaceutical prescribing, and pharmacists’ views on their new duties. Methods: The study used the method of a questionnaire, which was distributed to all pharmacists actively working in Poland. During the study, 309 completed questionnaires were obtained that gathered information about prescriptions written by pharmacists, as well as their opinions related to this entitlement. The results of the survey were analyzed using descriptive statistics. Results: Out of all authorized pharmacists, 75.62% use the new, extended authorization to issue pharmaceutical prescriptions. About half of them believe their prescriptions should be refundable. Only 11.52% of respondents do not use the entitlement to issue prescriptions for themselves or their family members. In addition, it was noticed that those who write such prescriptions most often use the fully paid version. Conclusions: Polish pharmacists use the new entitlements willingly but carefully. The legal provisions governing electronic prescription should be clearer. In addition, the idea of continued prescription should be developed as this is the most common reason that pharmacists issue prescriptions.

## 1. Introduction

Pharmacists are increasingly performing functions in health care systems attributed to independent medical professions. They have the knowledge and competence necessary to prescribe prescription-only drugs on their own. Until April 2020, a pharmaceutical prescription in Poland could be issued only in exceptional cases, as stipulated in Article 96 of the Pharmaceutical Law [1]. A pharmacist could dispense one minimum package size of a medicinal product admitted to marketing in the territory of Poland only in the case of an emergency threat to the patient’s health. Because of the uncertainty surrounding the definition of a medical emergency, and because such situations are rare in pharmacy practice, the decision to issue a pharmaceutical prescription was sporadic [2].

The specific types of prescribing among medical professions in Poland are *pro auctore* and *pro familiae*. A *pro auctore* prescription means a prescription for the prescriber, while a *pro familiae* prescription is issued to the family of the prescriber. The term “family” is used here to mean spouse, cohabiting partner, relations by blood and by affinity in the direct line, and relations by blood in the collateral line up to the third degree of kinship in Roman computation [1]. Doubts may be raised by the term cohabitation, which is not statutorily defined. *Pro auctore* and *pro familiae* prescriptions were the professional domain of physicians and dentists until nurses and midwives in Poland became eligible for such prescribing on 1 January 2016 [3]. In 2020, Polish pharmacists also became entitled to do so, with the coronavirus pandemic proving to be the turning point. The COVID-19 pandemic has put tremendous pressure on health care systems around the world. Difficult access to physicians has shown how fundamental all medical professions, including pharmacists, are to the functioning of the health care system. The widespread availability of pharmacies means that they are often a patient’s first point of contact with the health care provider. In the face of the pandemic, the need for additional pharmaceutical services, which required changes in legal regulations, was recognized in many countries, not only in Poland. For example, the implementation of e-prescribing in Italy and Austria, the possibility of the home delivery of medicines in Portugal and the USA or, as in Canada: authorizing pharmacists to prescribe, sell, or provide controlled substances, e.g., benzodiazepines, in limited circumstances, including allowing pharmacists to extend and renew prescriptions, as well as to transfer prescriptions to other pharmacists [4].

Currently, the entitlements of Polish pharmacists related to prescribing have been extended, with the introduction of legal regulations related to COVID-19 prevention in March 2020 [5]. Under this legislation, from 1 April 2020, a pharmacist has the ability to issue a prescription for any health risk to a patient [6]. The ongoing COVID-19 pandemic forces pharmacists to reinvent their roles and intensifies the pharmaceutical practice, whereas great shortages of the health care workforce these days make this initiative rational and helpful [7]. In light of the above-mentioned amendments, a medicinal product may be issued by a pharmacist in an amount sufficient for up to 180 days when an electronic prescription is used. As before, a pharmaceutical prescription is filled with a 100% payment. Medicinal products containing narcotic drugs and psychotropic substances are still excluded from pharmaceutical prescribing. A novelty in the entitlements of pharmacists is the possibility of issuing *pro auctore* and *pro familiae* prescriptions. Such a prescription may be reimbursable and may be issued by anyone in the pharmacy profession, not necessarily in a community pharmacy. Moreover, pharmacists are obliged to keep records of issued *pro auctore* and *pro familiae* prescriptions. The list of *pro auctore* and *pro familiae* prescriptions should contain information required by law: personal ID number, the diagnosis of the illness, health problem or injury, the International Nonproprietary Name (INN) or trade name of the medicine, the form and the dose of the prescribed drug, and the quantity. It is also very important to record the dosage regimen.

Based on data of the Ministry of Health collected between 1 June 2020 and 30 September 2020, the average number of prescriptions issued by pharmacists is gradually increasing—not only those for patients, but also for the pharmacists themselves and their families [2]. However, there are no data regarding what situations the prescriptions relate to—whether the majority of the prescribed medicines are those which are suddenly needed (life-saving drugs) or those used as part of a continuation of long-term therapy. The next stage of development of pharmaceutical prescribing in Poland will be a repeat prescription (from 16 January 2022) which will be possible for a pharmacist to issue within the scope of pharmaceutical care referred to in the Act of 10 December 2020 on the Pharmacist Profession [8]. 

Pharmaceutical prescribing is practiced worldwide based on different models. The basic division is independent prescribing (IP) or dependent prescribing. There is also collaborative prescribing, where the pharmacist’s prescribing is mainly based on the continuation of a doctor’s prescription [9]. An intermediate scheme between the models discussed above is supplementary prescribing (SP), which involves a voluntary partnership between a physician and a pharmacist (or nurse) to implement an individual clinical management plan (CMP) with patient consent. In this system, the pharmacist has access to the patient’s clinical data and may participate in their treatment process; moreover, the pharmacist may issue reimbursement prescriptions or modify the pharmacotherapy suggested by the physician. These solutions operate successfully in the United Kingdom [10]. Currently, in Poland there is no classification of pharmaceutical prescribing models; however, the current pharmacists’ entitlements are most similar to the idea of independent prescribing (IP). In light of the Act of 10 December 2020 on the Pharmacy Profession, there will also be a repeat prescription [8], so this change will presumably lead to collaborative prescribing, as we are still waiting for detailed guidelines on such prescriptions.

### Aim of the Study

Due to the extension of pharmacists’ prescribing rights in Poland, it was decided in this study to determine the current practice of pharmaceutical prescribing, and pharmacists’ views on their new duties. The study aimed to assess the opinions of Polish pharmacists on prescribing in the context of the legal changes, and their opinions on clarifying certain legal inconsistencies in this area. Moreover, it sought to define pharmacists’ concerns related to pharmaceutical prescribing.

## 2. Materials and Methods

### 2.1. Research Tools

Due to the lack of available surveys concerning pharmaceutical practices in the field of prescribing, and opinions of Polish pharmacists about the self-administration of prescriptions, a self-designed questionnaire was constructed based on articles referring to experiences with pharmaceutical prescribing in other countries [11,12,13]. The Survey questionnaire can be found at the Appendix A. In addition, publications referring to other health professionals, e.g., nurses, were used. The final form of the questionnaire was adjusted to realities resulting from the current legal status in Poland. To create the questionnaire, we used a content validity measurement: during the survey design, 5 scientists from the Sociology Department were asked to rate the survey questions and categorize them as “essential”, “useful”, or “not necessary”. For each question, the Content Validity Ratio was calculated, and the results ranged from 0.6 to 0.9.

The study was carried out by means of a diagnostic survey, using the questionnaire technique. The survey form consisted of 19 closed and semi-open questions. The first part of the questionnaire consisted of 15 questions and was designed to gather information about prescriptions written by pharmacists, as well as their opinions related to this entitlement. The first seven questions sought to determine whether pharmacists issued pharmaceutical prescriptions and *pro auctore/pro familiae* prescriptions—including whether they were reimbursement or fully paid prescriptions. Respondents were given a choice of one of three responses: “yes”, “no”, or “I do not remember”. The average number of prescriptions dispensed per month was then requested and the responses were given in numerical ranges. The construction of this question was based on a study conducted in Wales [13]. The respondents were then asked to specify what the prescriptions they had issued were for by indicating the main reason for doing so (continuation of therapy for a chronic disease, lifesaving, self-diagnosis, request from a patient or family member, other answer). A Likert scale (definitely no, rather no, hard to say, rather yes, definitely yes) was used in the question on the perception of current regulations as clear and transparent. The last six questions sought to uncover pharmacists’ opinions on the functionality of the current legal regulations, e.g., addressing the lack of reimbursement of pharmaceutical prescriptions. The next question raised the issue of magistral drugs in asking whether pharmacists should be allowed to write prescriptions for them. Respondents were given a choice of answers: “yes”, “no”, “I have no opinion”. The penultimate, 14th question related to feelings of anxiety about prescribing, and respondents who answered “yes” were asked in question 15 to indicate the reason for such feelings. The second part of the questionnaire contained 4 questions relating to the professional status of the respondents, their professional experience (with a choice of a range of years), the place of practicing their profession—with the possibility of indicating this place independently if it was other than a community or hospital pharmacy. The last question of this section asked about the location of the pharmacy or workplace by indicating the appropriate population density range of the locality.

### 2.2. Study Group

Pharmacists with a valid license to practice, and entered in the register of pharmacists, participated in the study; no exclusion criteria were applied, because all pharmacists with a license to practice may issue *pro auctore* and *pro familiae* prescriptions. Due to the narrow exclusion criteria, the snowball method was used to collect the data. It is a non-random sampling method that recruits participants by other participants. 

This method was able to obtain answers from 309 respondents, including 283 pharmacists working in community pharmacies, 6 master pharmacists from hospital pharmacies, and 20 pharmacists practicing in a location other than a pharmacy (hospice, etc.). Managers of community pharmacies accounted for 27.51% and other pharmacists working in community pharmacies 64.08% of all respondents, pharmacists working in hospital pharmacies accounted for 1.94%, while no responses were collected from managers of hospital pharmacies. Pharmacists practicing outside of a pharmacy accounted for 6.47% of the respondents. The most frequently indicated place of practice was a community pharmacy belonging to a large pharmacy chain, which concerned 211 respondents (74.55%). In total, 33 respondents came from small chain pharmacies, while 24 were from individual pharmacies.

### 2.3. Study Settings

The survey, using a questionnaire technique, was conducted among pharmacists from 3 January 2021 to 21 February 2021. The survey form was created electronically and distributed via email and social media, including being shared on a pharmacy staff internet group. Moreover, with the consent of the Dean of the Faculty of Pharmacy of the Medical University of Gdansk, the survey was conducted among doctoral students and academic teachers of this faculty. 

### 2.4. Ethical Issues

No intervention was performed on the research participants (non-invasive research) and therefore the consent of the bioethics committee was not required. The full confidentiality and anonymity of the respondents was maintained during the study. All participants gave their informed and voluntary consent prior to participation.

### 2.5. Statistical Analysis

Microsoft Excel 365 and Statistica 13.3 were used to process the results of the study. The statistical analysis was performed using Pearson’s chi square test. Differences of *p* < 0.05 were considered statistically significant.

## 3. Results

Respondents who reported professional experience of less than five years accounted for 39.16%. On the other hand, 28.48% of the responses are from pharmacists with professional experience ranging from 5 to 10 years and 27.18%, with experience ranging from 10 to 20 years. The fewest responses were received from respondents with work experience longer than 20 years (5.18%). 

The first question of the survey was to determine whether pharmacists, with the awareness of their extended rights in this area, use the possibility of prescribing drugs. Taking into account that pharmacists working in a hospital pharmacy or in a place other than a pharmacy do not have the possibility to issue a pharmaceutical prescription for patients, only the responses of pharmacists and managers working in community pharmacies (283 respondents) were included in the analysis. In this group, the response was 75.62% affirmative and 24.38% negative. Accordingly, almost half of the respondents (49.51%) believe that a pharmacist should be able to issue reimbursement prescriptions, 39.81% of the respondents have the opposite opinion, while 10.68% have no opinion on this issue (Table 1).

The vast majority of pharmacists surveyed issued a *pro auctore* prescription—83.5% (*n* = 258); 15.9% of respondents marked a negative answer (*n* = 49) and two respondents (0.6%) could not remember whether they issued such a prescription. Of the 258 respondents who wrote a prescription for themselves, 1 person skipped the questions regarding the payment of prescriptions, i.e., did not mark any of the answers in these questions. For the remaining group (*n* = 257), 81.3% (*n* = 209) issued a fully paid *pro auctore* prescription only, 4 people (1.6%) only a reimbursement one, and 44 respondents (17.1%) issued such a prescription in both the reimbursement and fully paid versions. Respondents were found to be significantly more likely to write a fully paid *pro auctore* prescription than a reimbursement prescription (*p* < 0.001).

In the case of *pro familiae* prescriptions, the same order of questions was followed—firstly, the respondent was asked whether he/she had issued the indicated prescription at all, and then whether these prescriptions included reimbursement and fully paid prescriptions, respectively. A *pro familiae* prescription was issued by 76.4% of the respondents (*n* = 236), while 23.3% of the respondents (*n* = 72) chose the opposite answer and one person marked the “do not remember” option. In order to calculate the percentage of fully paid and reimbursement prescriptions, a group of 233 respondents was used, since 2 respondents claimed not to have issued either fully paid or reimbursement prescriptions in the next two questions. In turn, one person, despite confirming the fact of issuing a *pro familiae* prescription, skipped the questions specifying the type of payment. In total, 8 respondents (3.4%) issued prescriptions only for reimbursed drugs and 172 pharmacists decided to write *pro familiae* prescriptions only with 100% payment. Study participants who issued *pro familiae* prescriptions of both payment types accounted for 22.8% of the group that used the given prescribing method (53 out of 233 individuals) (Table 2). Respondents were found to be significantly more likely to write a *pro familiae* prescription in the fully paid version than in the reimbursement version (*p* < 0.001).

Based on the results collected, it was found that 11.52% (*n* = 36) of the study participants did not write either a *pro auctore* or a *pro familiae* prescription. Nevertheless, a large number of respondents, i.e., 221 (71.52%), took advantage of their entitlement by prescribing both for themselves and their family members. 

To find out the extent of pharmaceutical prescribing, the study participants were asked to indicate the numerical range of prescriptions written per month. The largest number of respondents (*n* = 231) indicated a range between 1 and 5, while between 5 and 10 prescriptions per month were issued by 36 people and between 10 and 15 by 14 of the surveyed pharmacists. There were 2 responses each of “15–20” and “over 20”. The remaining respondents marked the option “0” (*n* = 24) (Table 3). No relationship was found between the number of prescriptions (including *pro auctore* and *pro familiae* prescriptions) dispensed and the length of the pharmacists’ work experience. In addition, there were no significant statistical differences between the number of prescriptions issued in the group of managers and masters of community pharmacies. The question about the reason for prescribing by the pharmacist, regardless of the type of prescription, revealed that most prescriptions were prescribed for the continuation of therapy for chronic diseases (*n* = 230). Prescriptions based on own diagnosis, in the case of a minor illness, were prescribed by 30 respondents, while the same number of people marked the option “at the request of the patient”. Only 14 respondents out of a total of 309 did not answer this question. No respondents marked the response “prescriptions for life-saving drugs (e.g., epinephrine, insulin)” to the question. Respondents were then asked to rate the current legislation. According to 44.7% of respondents, the current regulations are rather clear and transparent, while only 2.6% answered “definitely yes”. For 11.0% of the respondents, the current regulations are definitely not clear or transparent and 22.3% of the respondents marked “rather not” to this question. The remaining 19.4% of responses were “hard to say”. There were no significant statistical differences in terms of work experience (in years) or work status regarding this question. In the next question, respondents were asked to give their opinion on the prevalence of prescription abuse by pharmacists. There were 198 responses denying that such a phenomenon occurred (64.1%), and far fewer respondents chose an affirmative response—38 (12.3%). The answer “do not know” was marked by 73 respondents (23.6%) (Table 3).

In the conducted survey, respondents were asked if they felt anxious about the act of prescribing, including both pharmaceutical prescriptions and *pro auctore* and *pro familiae* prescriptions. More than half of the respondents, i.e., 60.2% (*n* = 186), had no such concerns, while 117 (37.9%) admitted to being concerned. The answer “I do not know” was marked by 6 respondents (1.9%). 

The final question was directed at respondents who confirmed feeling anxious about prescribing, regardless of the type of prescription. They were asked to indicate the reason for their anxiety and the total number of responses obtained in this case is 140. The most frequently cited reason was ambiguity in the current regulations (*n* = 78), followed by the possibility of harm to the patient as a result of poor judgment (*n* = 30). Concerns about control activities carried out by the National Health Fund and the Province Pharmaceutical Inspectorate were also indicated (9 responses). The lack of substantive preparation for independent prescribing fills 7 respondents with anxiety, and 3 fear conflicts with physicians. The remaining single responses placed by the respondents in the questionnaire are presented in the table below (Table 4).

## 4. Discussion

The COVID-19 pandemic revealed the need for a well-prepared, highly integrated health system that responds effectively to emergencies. Pharmacists have proven themselves in many roles that allow them to use their knowledge and expertise on the front lines of disasters and epidemics [14]. When restructuring health services to respond to the current public health crisis, it is essential that, following a review of existing services, the hitherto unrealized potential of pharmacists is fully realized [15]. Following the legal changes, the expanded entitlements regarding prescribing have been used in Poland for a relatively short period of time, so there is a lack of concrete research towards actual pharmacy practice in this area. This study is the first to address this topic. The survey yielded a relatively large number of responses in less than 2 months, which proves the interest of Polish pharmacists in the discussed topics. The study revealed the practical conditions of pharmaceutical prescription, as well as the concerns of the pharmacy community about the use of this tool.

In the group of respondents employed in community pharmacies, 75.6% had issued a prescription to a patient in the pharmacy, indicating that pharmacists mostly use this entitlement. Of the remaining respondents of this group (24.4%), it is not known for what reason the respondents chose not to write a prescription—perhaps some of them had not yet encountered a situation where a patient needed such a pharmaceutical service. A retrospective study using data collected by the Provincial Pharmaceutical Inspectorate, conducted in 2020, showed that during the 17 years of the previous legal status in force (2002–2019), only 2189 pharmaceutical prescriptions were issued in one particular province (842 pharmacies). After the introduction of legislation in 2020 for pharmacist prescribing practices in the study region, a steady increase in the number of prescriptions issued by pharmacists was observed. In the period from April to September 2020, over 18,500 pharmaceutical prescriptions were recorded in the same area [2]. 

One of the most common problems identified in relation to pharmaceutical prescribing is the lack of adequate funding from the public budget. The lack of coverage of this service by reimbursement rules was the main reason why pharmacists in the UK qualified to do so did not use this prescribing entitlement [11]. Despite general awareness of the benefits of providing pharmaceutical services, there are significant limitations on public coverage. In the United States, only pharmacist activities, such as drug reviews, that are related to conducting a Medication Therapy Management (MTM) program may be reimbursed. However, providing pharmaceutical consultations or interfering in the self-medication process of patients is not subject to financing. In Canada, on the other hand, pharmaceutical prescribing is covered by the reimbursement system in only two provinces [16]. However, according to the current legislation, pharmaceutical prescriptions cannot be reimbursed—therefore the respondents were asked whether they think a pharmacist should be able to have this option. Based on the obtained results, it is difficult to determine the prevailing attitude of pharmacists in this regard. Admittedly, nearly half (49.5%) of the respondents believe that such a possibility should be available, while 39.8% are of the opposite opinion and the remaining 10.7% have no opinion on this issue.

In the current state of the law in Poland, only *pro auctore* and *pro familiae* prescriptions can be issued on a reimbursement basis. However, also in the case of these prescriptions, pharmacists in Poland remain cautious when deciding on the payment of the prescription. 

The results of this study show that pharmacists are keen to make use of the permission granted to prescribe for themselves. Most of respondents (83.5%) issued a *pro auctore* prescription, and slightly less, (76.4%) issued a prescription for a family member. In addition, most of the respondents admitted to issuing both prescriptions (71.5%; *n* = 221). Most of these prescriptions were fully paid, despite the possibility of reimbursement. The propensity to prescribe reimbursed drugs is higher for *pro familiae* prescriptions than for *pro auctore* prescriptions (26.2% compared to 18.7%). 

The relatively low proportion of prescriptions including reimbursement may be due to fear of unfavorable interpretation by the controlling authorities (NFZ, WIF), which was pointed out by 7.8% of respondents in a further question regarding concerns (Table 3). The rules of drug reimbursement in Poland are quite complicated. The right to reimbursement of a given drug depends on the medical indication. The medical indication can be related to the setting of an appropriate diagnosis. Unfortunately, Polish pharmacists, despite extended prescription entitlements, do not have official diagnostic competences. 

Prescribing medicines for oneself and one’s relatives raises questions. On the one hand, this practice saves time and reduces the potential costs associated with consulting another specialist. On the other hand, the emotional connection leads to a greater risk of overlooking important symptoms and, consequently, making a misdiagnosis. As with all prescriptions, the pharmacist may be the first and last qualified health care professional with whom a consultation is possible when a *pro auctore* prescription is filled by a health care professional. When considering authorizing a pharmacist to prescribe medicines for themselves, the potential risk of abuse is increased for several reasons: general knowledge of drug products and the confidence in their own judgment that goes with this, easy access to most preparations in the pharmacy, and long hours that discourage the extra effort of visiting a physician [17]. In a New York City survey of pharmacists (*n* = 131), most admitted to prescribing for themselves, and the most common reasons included the belief that a doctor would prescribe the exact same drug. Despite the acknowledgement of the prevalence of this practice, prescribing prescription-only drugs without medical supervision has been considered unprofessional and illegal [18]. However, the data on prescriptions for the prescribers themselves are not strictly controlled and if there are any irregularities in this area, they are only resolved when complaints are recorded [19] or by a challenge to the reimbursement by the relevant institution, which in Poland is the National Health Fund (NFZ). 

Most of the surveyed pharmacists (74.8%) issue between 1 and 5 prescriptions a month on average. This result suggests that pharmacists are very cautious about their new rights. In the UK, pharmacists with newly acquired privileges also initially took a cautious approach to their prescribing role [20]. At the same time, such an attitude may indicate prudence and a strong sense of professional responsibility.

In the previous state of the law, pharmaceutical prescriptions in Poland could basically only apply to life-saving drugs. Interestingly, in the question identifying the most common reason for a pharmacist’s prescription, no one indicated such drugs in their response. The results show that pharmacists are most willing to prescribe a drug as a continuation of therapy (74.4% of respondents). Data from a survey conducted in 2020 show that for a period of 7 months (April–September 2020), the highest percentage of prescriptions were those used for diseases of the cardiovascular system, the gastrointestinal tract and metabolic disorders, the nervous system, and those used in dermatological diseases [2].

Abroad, repeat prescriptions are written by pharmacists very frequently—a system of repeat dispensing for certain chronic medicines exists in Australia, New Zealand, Canada, the Netherlands, and the UK [21]. Pharmacists with prescribing authority are able to achieve similar clinical outcomes as physicians, including blood pressure control, medication adherence, and patient satisfaction in terms of the quality of health care [22].

It should be emphasized that Polish pharmacists usually do not have access to the official medical records of the patient. However, a chronically ill patient’s need for access to appropriate medications is relatively easy to verify. The pharmacist may ask to see the previous doctor’s prescription; moreover, he may recognize the patient who constantly buys the given drugs in his pharmacy, and in the case of relatives (*pro familiae* prescription), he is aware of the treatment being provided on the basis of personal relations. At the time of the survey, the decision to prescribe repeat prescriptions was undoubtedly facilitated by a general awareness of the difficult access to physicians due to the epidemic situation and the fact that discontinuing long-term therapy is a health risk.

The dynamics of legal changes, not only in social, but also in professional functioning, may contribute to the difficult assimilation of new regulations. Less than half of the survey participants (47.3%) believe that the current pharmaceutical prescribing regulations are clear and transparent, while one-third of the respondents (33.3%) do not perceive the introduced regulations as understandable. This makes it impossible to determine how favorably Polish pharmacists feel about the content of the introduced legal changes. Furthermore, no significant differences were found in relation to experience, with expected different opinions of young versus more experienced pharmacists.

The profession of a pharmacist is a profession of public trust. As such, pharmacists are subject not only to criminal and civil liability, but also professional liability. In the context of the new entitlements, the majority (64.1%) of pharmacists interviewed do not believe that there is any abuse of their prescribing powers. The risk of over-reaping the benefits of looser regulations may be greater from chain pharmacy owners than from pharmacists themselves. An example of undesirable behavior was the controversial use of a pharmaceutical prescription as a means of advertising a pharmacy—which resulted in a fine of PLN 40,000 (around USD 10,000) being imposed by the Pomeranian Pharmaceutical Inspectorate (WIF) in Gdansk [23]. The advertising of a community pharmacy in Poland is not allowed [8]. Moreover, according to the Code of Ethics of a Pharmacist of the Republic of Poland, market mechanisms, social pressures, or administrative requirements do not exempt a pharmacist from observing the principles of professional ethics [24].

Based on the experiences of UK and Canadian pharmacists, the greatest difficulties in making prescribing decisions included lack of confidence, risk of error, limited access to the full clinical picture, lack of prescribing experience, and in addition, inadequate levels of communication with patients as well as physicians. Barriers to pharmacist prescribing also include lack of proper procedures for managing pharmaceutical prescriptions, lack of subsidies for prescribing, potential conflict with other pharmacist duties, and interference in a professional area dominated by physicians [11]. Pharmacists have been shown to be confident in their independent prescribing [13]. In contrast, physicians were less favorable towards assigning such a function to pharmacists, with inadequate preparation in the area of clinical trials mainly indicated [25].

In this conducted survey, 37.9% of the respondents admitted that they felt apprehensive about their prescriptions. Among the most frequently indicated concerns were ambiguities in the current regulations (55.7%), which may result from the fact that the new entitlements were granted relatively recently (01.04.2020) and only from 15.05.2020 was the application enabling the issuance of e-prescriptions launched. This was followed by respondents who marked the possibility of harm to the patient as a result of misjudgment (21.4%), which may be due to a lack of appropriate professional training. Moreover, several respondents felt more than one concern regarding prescribing (Table 3). 

There are several limitations in this study that may affect the objectivity and reliability of the results obtained. The group of 308 surveyed pharmacists accounted for 0.85% of the total number of people meeting the survey criteria. Currently, 36,482 pharmacists with an active license to practice are entered in the register of pharmacists. To achieve the sample size recommended by the Sample Size Calculator, made available by the platform http://www.raosoft.com/ [26], the study group should have at least 381 respondents (confidence level: 95%, margin of error: 5%). The current number of respondents increases the margin of error to 5.56%. In addition, the form of only online distribution of the questionnaire did not allow any doubts related to the questions to be dispelled, as well as limiting the group of potential respondents to people actively using the Internet.

## 5. Conclusions

The amendment of the regulations on pharmaceutical prescription undoubtedly brings benefits both to patients and pharmacists. The privilege of prescribing raises the profile of the profession while providing a vehicle for pharmaceutical care. By elevating the role of pharmacists in the health care system, it is possible to increase the fluidity of health care services while relieving other medical professions. The results of the survey showed that pharmacists in Poland willingly but cautiously use the extended prescribing entitlements. The main reason for this is that the existing regulations are not fully clear to prescribers, therefore they should be made more specific and uniform. The most common reason identified by pharmacists for prescribing was the continuation of therapy, so a further development of the rights discussed could be a repeat prescription. Nonetheless, further research is needed to completely explore the realities and benefits regarding the expertise of prescribing pharmacists.

## Figures and Tables

**Table 1 ijerph-19-01648-t001:** Percentage distribution of respondents’ answers to the questions asked in the questionnaire regarding a pharmaceutical prescription issued to a patient in a pharmacy.

Question Asked in the Survey	Answer	All Respondents (*n* = 309)	Pharmacists and Managers Working in Community Pharmacies (*n* = 283)
Have you issued a pharmaceutical prescription (fully paid to the patient) after the legal changes?	Yes	71.52%	75.62%
	No	28.48%	24.38%
**Total:**		100%	100%
Should pharmacists be able to issue reimbursement prescriptions for patients?	Yes	49.51%	47.7%
	No	39.81%	42.05%
	Have no opinion	10.68%	10.25%
**Total:**		100%	100%

**Table 2 ijerph-19-01648-t002:** Distribution of respondents’ answers to questions about the *pro auctore* and *pro familiae* prescriptions issued.

Question Asked in the Survey	Answer	%	Frequency
Have you been issuing *pro auctore* prescriptions?	Yes	83.5%	258
	No	15.9%	49
	I do not remember	0.6%	2
**Total:**		100%	309
With what payment method do you issue *pro auctore* prescriptions?	Fully paid only	81.3%	209
	Reimbursement only	1.6%	4
	Both fully paid and reimbursement	17.1%	44
**Total:**		100%	257
Have you been issuing *pro familiae* prescriptions?	Yes	76.4%	236
	No	23.35	72
	I do not remember	0.4%	1
**Total:**		100%	309
With what payment method do you issue *pro familiae* prescriptions?	Fully paid only	72.9%	172
	Reimbursement only	3.4%	8
	Both fully paid and reimbursement	22.5%	53
**Total:**		100%	233

**Table 3 ijerph-19-01648-t003:** Distribution of respondents’ responses to questions about opinions and experiences with prescribing.

Question Asked in the Survey	Answer	%	Frequency
How many prescriptions do you issue on average in a month?	0	7.77%	24
	1–5	74.75%	231
	5–10	11.65%	36
	10–15	4.53%	14
	15–20	0.65%	2
	>20	0.65%	2
**Total:**		100%	309
What was the most common reason for prescribing?	Continuation of chronic diseases therapy	77.9%	230
	Based on own diagnosis, in the case of a minor illness	10.2%	30
	At the request of the patient	10.2%	30
	Other	1.7%	5
**Total:**		100%	
Do you think the current rules governing pharmaceutical prescriptions are clear and transparent?	Definitely yes	2.6%	8
	Rather yes	44.7%	138
	Difficult to say	19.4%	60
	Rather not	22.3%	69
	Definitely not	11.0%	34
**Total:**		100%	309
Do you feel that there is abuse of prescriptions by pharmacists?	Yes	12.3%	38
	No	64.1%	198
	I do not know	23.6%	73
**Total:**		100%	309

**Table 4 ijerph-19-01648-t004:** Distribution of responses to the question about what the perceived concerns about prescribing are.

What are the Concerns about Pharmacist Prescribing?	Number of Responses	Percentage of Responses
Lack of substantive preparation	7	5.0%
Conflicts with doctors	3	2.1%
The potential for harm to the patient as a result of poor judgment	30	21.4%
Ambiguities in the current legislation	78	55.7%
Control by NFZ */WIF **	9	6.4%
All the answers	2	1.4%
Abuse of prescriptions by patients	3	2.1%
My assessment of whether a lack of medication is a health risk versus the assessment of a possible controller	1	0.7%
Lack of a review of the prescribed medication history	2	1.4%
No remuneration for prescribing	1	0.7%
Other answer (combinations of several of the above)	4	2.9%
Total	140	100.0%

* NFZ (Narodowy Fundusz Zdrowia)—National Health Fund—the Polish state organizational department which is responsible for financing health care services. ** WIF (Wojewódzki Inspektorat Farmaceutyczny)—Provincial Pharmaceutical Inspectorate—regional unit of the state institution supervising control over the manufacturing and marketing of medicinal products in Poland to ensure patient safety.

## Data Availability

Not applicable.

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
