# Peer review of "Extended Prescribing Roles for Pharmacists in Poland—A Survey Study"

_ijerph, 2022, doi:10.3390/ijerph19031648_

Round 1

Reviewer 1 Report

1. Line 465 - is role of pharmacies, I propose - role of pharmacists.

2. Position "6" on references - lack of citation.

3. In the introduction, please refer to more Polish publications, i.e. Acta Poloniae Pharmaceutica – Drug Research, 2021, Vol. 78 No. 2 pp. 151–156.

Author Response

Thanks for all comments and remarks. The manuscript has been revised as suggested.

  1. Line 465 - is role of pharmacies, I propose - role of pharmacists.
    1. Corrected as suggested
  2. Position "6" on references - lack of citation
    1. The missing designation was completed
  3. In the introduction, please refer to more Polish publications, i.e. Acta Poloniae Pharmaceutica – Drug Research, 2021, Vol. 78 No. 2 pp. 151–156.
    1. Corrected as suggested

Reviewer 2 Report

Kindly do the minor changes as highlighted in the attached file. 

Author Response

Thanks for all comments and remarks. The manuscript has been revised as suggested.

Reviewer 3 Report

This original research paper is an important report that presents the results of a questionnaire survey on the role of pharmacists in extended prescribing in Poland with necessary information. The author provided enough information, however, please specify the number of pharmacists surveyed and the response rate to indicate the scale of the survey.

Author Response

Thanks for all comments and remarks. The manuscript has been revised as suggested.

  1. Please specify the number of pharmacists surveyed and the response rate to indicate the scale of the survey.
    ANSWER:
    In the "methods" section, more details about the method of recruiting respondents to the survey have been added.